# Estimating the effect of tracking tag weight on insect movement using video analysis: A case study with a flightless orthopteran

Oto Kaláb[1], David Musiolek[2], Pavel Rusnok[3], Petr Hurtik[3], Martin Tomis[4], Petr Kočárek[1] *

1 Department of Biology and Ecology, Faculty of Science, University of Ostrava, Ostrava, Czechia, 2 Faculty of Forestry and Wood Sciences, Czech University of Life Sciences Prague, Prague, Czechia, 3 Institute for Research and Applications of Fuzzy Modeling, Centre of Excellence IT4Innovations, University of Ostrava, Ostrava, Czechia, 4 Faculty of Electrical Engineering and Computer Science, VSB - Technical University of Ostrava, Ostrava, Czechia

* petr.kocarek@osu.cz

**Data Availability Statement:** All data (video frames, tracking data and tables with experiment data) are available from the Zenodo repository (DOI 10.5281/zenodo.4498323). The tracking software

## Abstract

In this study, we describe an inexpensive and rapid method of using video analysis and identity tracking to measure the effects of tag weight on insect movement. In a laboratory experiment, we assessed the tag weight and associated context-dependent effects on movement, choosing temperature as a factor known to affect insect movement and behavior. We recorded the movements of groups of flightless adult crickets *Gryllus locorojo* (Orthoptera: Gryllidae) as affected by no tag (control); by light, medium, or heavy tags (198.7, 549.2, and 758.6 mg, respectively); and by low, intermediate, or high temperatures (19.5, 24.0, and 28.3°C, respectively). Each individual in each group was weighed before recording and was recorded for 3 consecutive days. The mean (± SD) tag mass expressed as a percentage of body mass before the first recording was 26.8 ± 3.7% with light tags, 72 ± 11.2% with medium tags, and 101.9 ± 13.5% with heavy tags. We found that the influence of tag weight strongly depended on temperature, and that the negative effects on movement generally increased with tag weight. At the low temperature, nearly all movement properties were negatively influenced. At the intermediate and high temperatures, the light and medium tags did not affect any of the movement properties. The continuous 3-day tag load reduced the average movement speed only for crickets with heavy tags. Based on our results, we recommend that researchers consider or investigate the possible effects of tags before conducting any experiment with tags in order to avoid obtaining biased results.

## 1 Introduction

Researchers have attached small devices to mobile insects for various purposes. In ecological research, the devices are usually used to track animal movement, space use, and behaviour [1, 2]. For the tracking of small animals like insects, "tags" are typically attached to the animal's body. These tags can be passive (with radio-frequency identification (RFID) or harmonic

build and source code is available at https://doi.org/10.5281/zenodo.5081953.

**Funding:** Funding was provided by a University of Ostrava student grant (SGS14/PrF/2018) and the Orthopterist's Society by The Theodore J. Cohn Research Fund.

**Competing interests:** The authors have declared that no competing interests exist.

radar methods) or active (with the radio telemetry method), and each method has its significance in research and is suitable for different types of studies [1, 3]. One factor that researchers must consider when choosing a tracking method is the weight of the tag with respect to ability of the studied species to carry additional weight. Tag weights range from less than 1 mg (passive tags) to hundreds of mg (active tags). Among the above-mentioned methods, studies using active radio telemetry tags have had the highest tag/body mass ratio; in some cases, the tag weighed more than the animal itself [3]. Light radio telemetry tags (e.g., those weighing 200 mg) can be used on a large number of animal species including flying insects [4–6], but their batteries only function for a few days (up to ca. 20 days depending on pulse rate and length configuration), and they are therefore suitable only for short-term studies [7]. Heavier radio telemetry tags, in contrast, last longer than lighter tags (up to ca. 45 days with a 500-mg tag depending on pulse rate and length configuration) but their use is limited to relatively large species of insects, such as *Anabrus simplex* Haldeman, 1852 [8] or *Deinacrida heteracantha* A. White, 1842 [9], which have the ability to move with additional weights. Thanks to technological advances, the size and weight of telemetry tags have decreased over time [10, 11], and telemetry tags are increasingly being used to study insect ecology and evolution [1, 12, 13].

Despite the increasing use of tracking in the study of insects, only a few insect tracking studies have quantified the effect of the tag attachment [3, 6]. Some studies performed additional tests of tag attachment effects (e.g., [8]), while others examined the effect *post hoc* from field data (e.g., [9]). The majority of studies, however, only discuss the possible effects of tags [3], and some studies simply ignore them (e.g., [14]). In addition to potentially affecting movement, tags may also have energetic costs [1], which may result in body mass changes [15] and increases in resting periods [5]. These effects of tags can differ depending on the attachment duration [5, 16] and external conditions (stressors, environment) [3, 17]. Because the effects of tag attachment are species-specific and context-dependent [3, 18, 19], researchers have recently indicated that there are no general rules regarding the best tag/body mass ratios, i.e., the effect of the tag on the results should be considered for each specific study [3]. Failing to account for possible tag effects can lead to biased movement data and thus to inaccurate interpretation of results in ecological, conservation, or pest control studies [3]. In addition to being fitted with tags for movement tracking, insects can be fitted with devices for other purposes such as IoT sensing [20] or video recording [21]. It follows that an understanding of the effect of tag attachment is needed to avoid obtaining unreliable results in various fields. In contrast to the limited data concerning the effects of tracking tags on arthropods, there are substantial data concerning the effects of tracking tags on vertebrates [22–26].

There are, for example, known context-dependent effects of tag attachments on birds, causing researchers to conclude that temperature, breeding stage, and brood size should be considered in tagging studies [26]. Temperature greatly influences insect movement [27, 28] which usually decreases under suboptimal thermal conditions [29], because insect muscular efficiency and power output is highly temperature-dependent [28]. Although we are unaware of any insect study that quantified how the effects of tag attachment are altered by temperature, the importance of environmental and stressor factors when tagging arthropods was recently discussed in a review by Batsleer et al. [3]. The lack of knowledge about the effect of temperature on how extra tag weight affects arthropod movement may bias results from some field studies.

One way to evaluate the effect of tag attachment on insect movement is to use image/video analysis for tracking. Tracking software is commonly used in various experiments [30, 31] and behavioural studies [32–34], but can also be used for animal tracking as an alternative to the use of RFID tags [35]. A few previous studies used video recording to estimate the effect of tag attachment on insect movement and behaviour. Lorch et al. [8] studied the effect of radio

transmitter attachment on the movement of the *Anabrus simplex*; they recorded individuals in a 26.6-cm-diameter circular arena and analysed the data with *Noldus EthoVision* software [36]. Another series of studies [16, 18, 30, 37] assessed the effect of harmonic radar tag attachment on the survival, behaviour, and movement of insect pests. Boiteau et al. [18] studied the effects of glues used to attach tags, and Boiteau et al. [37] studied the effects of tag weight on three Coleoptera species. Lee et al. [16] studied the effects of both glues and tag weight on *Halyomorpha halys* (Stål, 1855) (Heteroptera) adults, and Kirkpatrick et al. [30] studied the effects of glues and tag weight on *H. halys* nymphs. All of these studies used a similar methodology. The insects were individually recorded under fixed environmental conditions in 10-cm-diameter Petri dishes, and recordings were analysed with *Noldus EthoVision* software [36], except that Boiteau et al. [37] used a 70-cm diameter wooden arena for one of the studied species and analysed movement with a *Videomex V* motion monitoring device (Columbus Instruments, Columbus, OH).

In the current study, we describe an inexpensive and rapid method to assess the influence of tag attachment on small, ground-dwelling, flightless animals. The described method uses 3D-printed dummy tags and video analysis with identity tracking. Our method differs from the previous studies in that we used identity tracking software that enabled us to track multiple individuals from one record acquired at the same time from the same arena. By allowing researchers to obtain multiple individual trajectories from a single video record, this approach reduces the time required for data collection and subsequent video analysis, Moreover, these trajectories are obtained under homogenous conditions. As a case study, we conducted a laboratory experiment using the cricket *Gryllus locorojo* Weissman & Gray, 2012 (Orthoptera: Gryllidae) as a model species. Based on previously published research concerning tag attachment effects, we attempted to answer the following four questions: (i) How does tag weights affect cricket movement? (ii) Do the effects of added weight on cricket movement change with temperature? (iii) Do the effects of added weight change during a 3-day period? and (iv) How does the added weight affect cricket body mass over time?

## 2 Materials and methods

### 2.1 Model species

We used the crazy red field cricket *Gryllus locorojo* as a model species. This species is widely distributed by commercial pet food farms (in Europe, it is misidentified as a *Gryllus assimilis* (F.)) [38]. The area of origin of this species is unclear, but it could be Ecuador [38, 39]. The biology of *G. locorojo* is also not well known. Previous experiments with *G. locorojo* were conducted with temperatures ranging from ca. 23 to 27˚C [40–42], and it can be raised at temperatures from 28 to 32˚C [38]. *G. locorojo* was chosen because it is a medium-sized orthopteran and is flightless, easy to obtain, and has sufficient size for the experiment; the average weight (± SD) of individuals used in our study was 768 ± 117 mg, which enabled us to obtain a wide range (from 18 to 127%) of tag masses expressed as percentages of cricket body mass. A culture of the cricket was obtained from a local pet food supplier (ACHETA farm, Czech Republic).

After the crickets were obtained from the supplier, they were reared in well-ventilated plastic containers (40 × 30 × 20 cm), with up to 100 nymphs or 50 adults per container. The containers were kept in a laboratory with constant temperature 22 ± 2˚C and a 14:10h light:dark photoperiod. Water and food were provided *ad libitum*. The food consisted of mixture of dry cat food and fish food as in Booth & Kiddell [43]. The containers were equipped with egg cartons as shelters and plastic containers filled with a moist sand-peat mixture for egg-laying. Relative uniformity in cricket size, health and performance was assured by using second-generation laboratory-reared individuals for the experiment. To avoid the effect of sexual

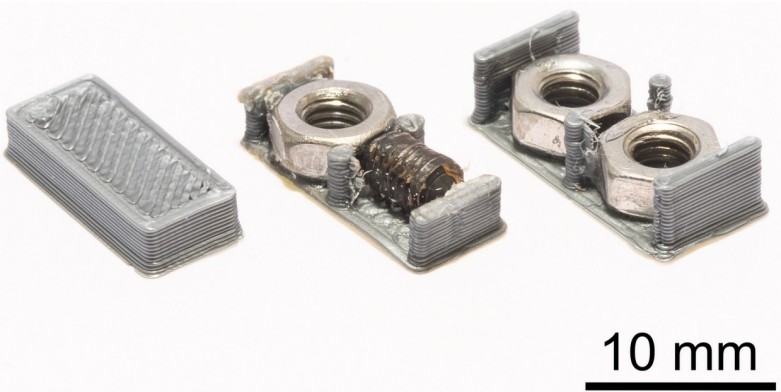

**Fig 1. Light, medium, and heavy (from left to right) dummy tags constructed based on the weights and dimensions of three commonly used radio transmitters.**

behaviour (courtship, phonotaxis) and senescence, only 7- to 14-day-old post-imaginal molt virgin females were used.

## 2.2 Dummy tags

We designed three types of dummy tags: light (mean mass ± SE) = 198.7 ± 2.1 mg; medium (549.2 ± 7.3 mg); and heavy (758.6 ± 6.9 mg) (Fig 1). The masses of these dummy tags were similar to those of commonly used, commercially available radio transmitters (Table 1). We used a 3D printer with a PETG filament to create these tags. The light tag was printed as a compact brick, and the medium and heavy tags had a printed base to which was glued headless screws (M3 × 5 socket set screws) and nuts (M3 A2 stainless steel nuts) to achieve the desired weight (Fig 1). The tags were designed without wire antennae to avoid antennae interacting with the experimental arena because the variable of interest was tag weight. We constructed 45 dummy tags of each type. The tags were glued (Loctite ethyl 2-cyanoacrylate adhesive) to the cricket pronotum. With attachment to crickets, the mean (± SD) tag mass expressed as a percentage of body mass before the first recording was 26.8 ± 3.7% with light tags, 72.0 ± 11.2% with medium tags, and 101.9 ± 13.5% with heavy tags.

**Table 1. Dimensions and weights of the three small radio telemetry tags produced by various manufacturers (ATS, lotek, and holohil) and of the 3D printed dummy tags used in the current study.**

| Manufacturer | Tag model | Dimensions (mm) | Weight (mg) |
|---|---|---|---|
| ATS | A2412 | 12 × 5 × 1.5 | 200 |
| | A2415 | 13 × 5 × 3 | 500 |
| | A2435 | 14 × 6 × 4 | 750 |
| Lotek | PicoPip (Ag190) | 12 × 5 × 2 | 200 |
| | PicoPip (Ag376) | 17 × 8 × 5 | 560 |
| | Pip (Ag376) | 17 × 8 × 5 | 700 |
| Holohil | LB-2X | 8 × 4 × 2.8 | 270 or 310 |
| | LB-2 | 13 ×6.5× 3.5 | 470 or 520 |
| | BD-2 | 14 × 6.5 × 3.5 | 620 or 750 |
| our dummy | light | 11 × 5 × 2 | 198.7 (SD ± 2.1) |
| | medium | 12 × 5 × 3 | 549.2 (SD ± 7.3) |
| | heavy | 14 × 6 × 4 | 758.6 (SD ± 6.9) |

## 2.3 Arena

For the experiment, we built a single 1.2 × 0.8 × 0.4 m (L × D × H) open-top glass arena in the laboratory. The inner bottom surface was covered with a black, rough cotton fabric to provide traction for cricket movement. The side walls of the arena were covered (on the outside) with white polystyrene plates to provide homogenous light distribution and to block outside visual cues. There were no shelters or obstacles except a temperature probe for data logging (TMP-BTA, Vernier Software & Technology, USA), which was located in the centre of the arena floor. The arena was lit with a UV black light bulb (25 W power, wavelength 368 nm) situated centrally, 1.5 m above the arena floor. The temperature in the arena was regulated by the combination of air conditioning in the laboratory and a floor heating foil (power 80W/m$^2$; Fenix s.r.o., Czech Republic) situated under the bottom of the arena.

## 2.4 Experimental design

In total, 180 haphazardly chosen individuals were divided into 3 groups that were studied at a low temperature (19.5˚C ± 0.4), 3 groups that were studied at an intermediate temperature (24.0˚C ± 0.4), and 3 groups that were studied at a high temperature (28.3˚C ± 0.3). Each group consisted of 20 individuals included 5 control individuals (without tags) and 5 individuals for each tag weight (light, medium, and heavy). Each individual within one group was labelled with a unique mark using a UV active colour with one of 10 symbols (*Z, X, B, O, M, H, 4, K, V, +*) in green or red to achieve 20 unique markings in order to unambiguously distinguish between them (Fig 2). Each dummy tag was marked and weighed to the nearest mg before the experiment. The control individuals were marked with a piece of paper attached to the pronotum in the same way as individuals with dummy tags.

The dummy tags and labels were attached 3–6 h before the first recording of the group, and the crickets were kept in the breeding boxes until recording. At 1 to 3 h after the beginning of the dark photoperiod, each individual in a group was weighed (to the nearest mg) and placed in the arena, which was lit only by the UV black light. After a 15-min habituation, we began a 10-min recording (top view) of the movement of the 20 crickets with a 4K camera (Panasonic DMC-G80, with Panasonic Leica DG Vario-Elmarit 12–60mm f/2.8–4 Power O.I.S. lens @

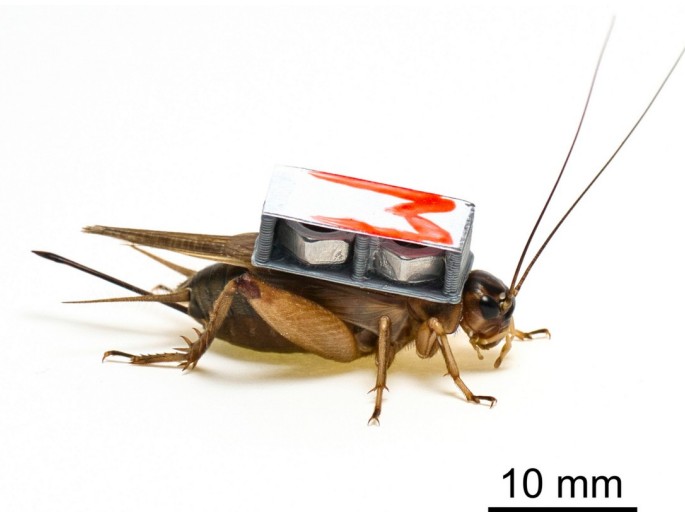

**Fig 2. Attachment of the heavy dummy transmitter with a unique UV paper label on a *Gryllus locorojo* female.** Only UV paper labels were attached to control crickets.

**Table 2. Data captured during the experiment.**

| Variable | Description |
|---|---|
| temperature | Three categories (low, intermediate, high) based on average temperature (in ˚C) during the 10 minutes of video recording. |
| day | Day of continual attachment duration (Day 1, Day 2, Day 3). |
| weight category | Attachment weight categories (control, light, medium, heavy). |
| tag weight | Exact weight of attached dummy in milligrams. |
| animal weight | Exact weight of every cricket in milligrams weighted before each recording. |

12mm, f/2.8, ISO 1600, SS 1/50 s, 24 fps, 3840 × 2160 px). The temperature was measured continuously during the habituation and video recording, and averages (± SD) were used to represent the temperature categories. The recording procedure using the same cricket group was repeated on 3 consecutive days to evaluate the long-term effect of the tag attachment. All captured variables are summarised in Table 2.

For an illustration of video recordings, see our demonstration movie on YouTube (https://youtu.be/PYydVG6gjE0). The high-resolution output with 20 moving individuals contains substantial information and requires substantial processing power. Therefore, a 10-min recording was considered sufficient for obtaining a useful dataset. After the recording was completed, the crickets were immediately returned to the breeding boxes.

In summary, there were three replicate groups for each of the 3 temperatures, while each single group was recorded in 3 consecutive days. The same crickets were used for the same temperature treatment on each day. We captured a total of 27 video records (Fig 3).

## 2.5 Transforming visual data into numerical data

For the transformation of visual data into numerical data, we obtained the recorded movies as described in section 2.4, processed the data with tracking software, and obtained a movement trajectory for each individual insect. The movement trajectory was then used to compute the characteristics that are evaluated in the Results.

Our original idea was to use an existing video tracking application. Unfortunately, the best of the available applications for animal tracking such as *Ctrax* [44], *SwisTrack* [45], and *Winanalyze* (https://winanalyze.com) failed to handle our 4K resolution recordings. We could not use other well-known applications, including *Bio-Records* (http://www.bio-tracking.org) or *Noldus EthoVision* [36] because they do not provide "identity tracking", which means they can swap identification of individuals and therefore invalidate the results. By "identity tracking", we mean a process in which multiple individuals are tracked without any swapping of identities among individuals. Other potentially useful applications that do enable identity tracking have been presented in scientific papers, and these include *Tracktor* [46], which is based on adaptive thresholding, and *PathTrackrR* [47], which utilizes its own tracking core. Unfortunately, *Tracktor* cannot handle occlusion (the blocking of one object by another), and *PathTrackrR* can track only one object. We, therefore, developed our own application that enables the identity tracking of insects [48]. After we developed and published our algorithm, three other interesting approaches were published, i.e., *ToxTrac* [49], *GRAPHITE* [35], and *TRex* [50]. *ToxTrac* is fast and simple, but we found that it generated new IDs if occlusion occurs (unpublished observations), which was not acceptable for our experiment. Furthermore, *ToxTrac* is unable to produce statistics for each individual insect. *GRAPHITE* can recognize tags on insects and can therefore be used for identity tracking with a low tracking error. Because *GRAPHITE* is written

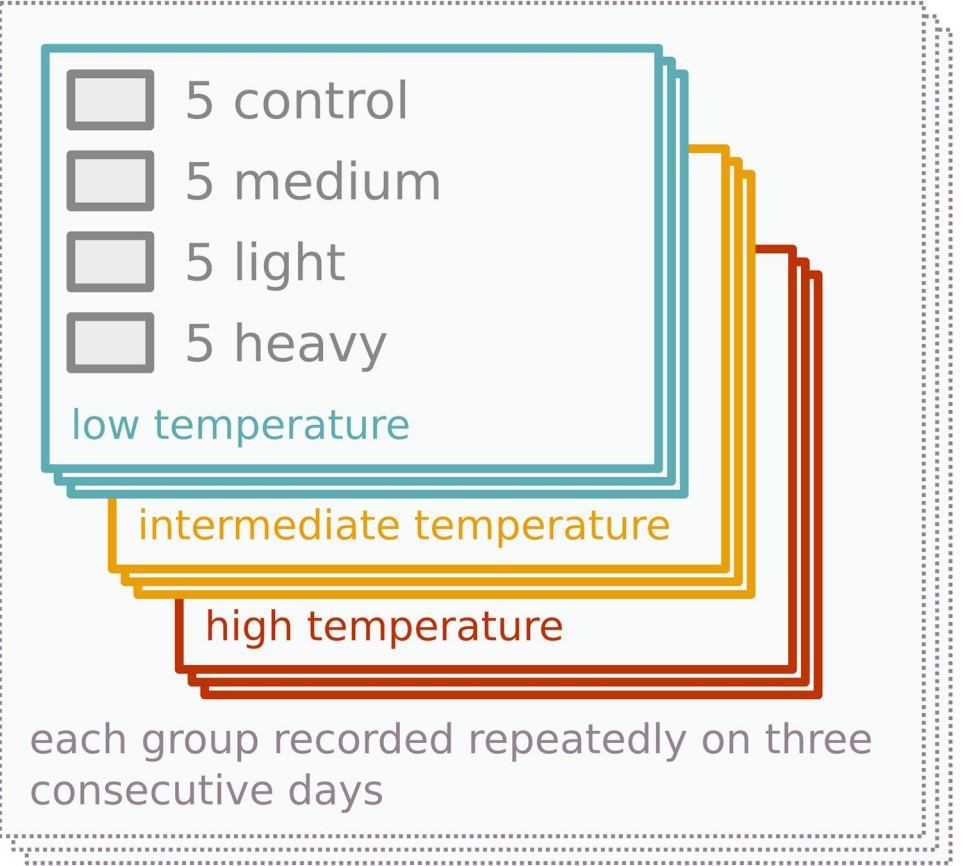

**Fig 3. Experimental design.** There were 5 control individuals and 5 individuals for each tag weight category in each group. Movement of each group at one of three temperatures (19.5, 24.0, or 28.3˚C) was recorded repeatedly on 3 consecutive days.

in Matlab and requires several nonfree toolboxes, we were unable to test it. *TRex* is a free, open source, modern, and complex software offering connection to multiple video sources, tracking functionality, and computation of statistics. These authors report the software is also the fastest one. *TRex* however, requires an NVidia graphics card because it utilizes deep neural networks. Without that, the application speed is reduced by an approximate factor of 20.

We therefore used a custom software developed by our team; the entire algorithm, including benchmark and difficulties, is described by Hurtík et al. [48]. The application's build and source code is available at https://doi.org/10.5281/zenodo.5081953. As indicated earlier, our goal was to measure the characteristics of multiple individuals in a group, and swapping between two individuals would not provide the required results. With the custom software, we were able to conduct identity tracking, i.e., we were able to determine the movement trajectories of individuals in a group of 20 individuals. Being able to assess movements of an individual in a group was considered important because it accelerates data acquisition and eliminates the effects of stochastic environmental variation experienced by crickets in a treatment group.

## 2.6 Data processing and statistical analysis

With the conversion of visual data into numerical data, we obtained spatio-temporal movement data (e.g., *x*, *y* positions in each video frame) of each cricket. To analyse the relationships

**Table 3. Movement and resting properties that were calculated from the data.**

| Property name | Definition | Unit |
|---|---|---|
| movementLength | The average length of all signle *movements*. | m |
| movementLengthMax | The maximal length of all signle *movements*. | m |
| movementSum | The sum of lengths of all *movements* (overall travelled distance). | m |
| movementSpeed | The average speed calculated on all *movements*. | m/s |
| restingDuration | The average *resting* duration. | s |
| restingFrequency | The total count of *resting* periods. | |

among variables (Table 2) and movement data, we calculated various movement properties for each cricket. First, we defined two main types of movement behaviour: *movement* and *resting*. Only continuous movement that was longer than 2 cm was considered to be *movement*. This approach eliminates noise in the data caused by very small movements. *Resting* occurs when the cricket stops and remains still or performs very small movements $\leq$ 1 cm/s. Second, we calculated six movement properties: *movementLength*, *movementLengthMax*, *movementSum*, *movementSpeed*, *restingDuration*, and *restingFrequency* (summarized in Table 3).

To investigate the effects of tags on movement properties, we compared each group of crickets with attached dummy tags (light, medium, or heavy) to the group without tags (control). We performed two sets of tests: one in which temperature was not included as a source of variance to simulate the long-term natural condition, and the other with separate temperature groups (low, intermediate, and high) to estimate the influence of single temperature levels. We performed Shapiro-Wilks test on each movement property on each weight category on each day to decide whether to use parametric or non-parametric tests. Some of the measurements did not have a normal distribution (the p-value of Shapiro-Wilks test was < 0.05). We therefore used the non-parametric Mann-Whitney test to determine whether tag weight or temperature affected movement properties. To reduce false positive significance (type I error) due to performing multiple tests, the p-values of Mann-Whitney tests were adjusted according to Yekutieli & Benjamini [51]. We also used mixed linear models to quantify the influence of the tag weight on distance travelled (*movementSum*). Mixed linear models were defined based on the experiment structure and two-way analysis of variance. We arrived at the following model: *movementSum* $\sim$ *temperature* $*$ *tag weight* + *temperature*$*$*animal weight* + (1|*idCricket*), where *idCricket* is a random effect of repeated measuring of a cricket.

We also used the Wilcoxon paired test to determine how cricket body mass changed during the 3 days of attachment and simple linear models to determine the effect of attachment duration on movement properties. The statistical analyses were performed with R software [52], with *base R* functions for paired tests and with functions from the *lme4* [53] package for linear mixed-effects models. Other possible analysis options such as the use of fuzzy logic are described in our preliminary analysis of the data [54].

## 3 Results

### 3.1 Movement properties

Mann-Whitney tests showed that the light tag significantly decreased the total distance travelled (*movementSum*) on day 2 ($p < 0.05$), the average movement speed (*movementSpeed*) on day 1 and 2 ($p < 0.05$), and number of stops (*restingFrequency*) on day 1 and 3 ($p < 0.05$). The medium tag significantly decreased the average maximum length of movement (*movementLengthMax*) on day 1 and 2 ($p < 0.05$), the total distance travelled (*movementSum*), and the average movement speed (*movementSpeed*) on all 3 days ($p < 0.05$), and increased the resting

**Table 4. Medians of movement properties of control crickets and crickets carrying light, medium, or heavy tags without regard to temperature.** Gray highlighted values indicate significant Mann-Whitney tests ($p < 0.05$) for comparisons of movement properties of crickets with and without tags.

|  | control | light | medium | heavy |
|---|---|---|---|---|
| Day 1-movementLength | 0.075 | 0.070 | 0.071 | 0.065 |
| Day 2-movementLength | 0.078 | 0.070 | 0.073 | 0.072 |
| Day 3-movementLength | 0.080 | 0.077 | 0.076 | 0.068 |
| Day 1-movementLengthMax | 0.386 | 0.305 | 0.289 | 0.273 |
| Day 2-movementLengthMax | 0.384 | 0.311 | 0.276 | 0.287 |
| Day 3-movementLengthMax | 0.406 | 0.385 | 0.317 | 0.297 |
| Day 1-movementSum | 15.63 | 12.21 | 11.79 | 9.19 |
| Day 2-movementSum | 19.84 | 16.08 | 15.66 | 12.04 |
| Day 3-movementSum | 18.97 | 17.63 | 13.34 | 11.73 |
| Day 1-movementSpeed | 0.064 | 0.057 | 0.056 | 0.051 |
| Day 2-movementSpeed | 0.066 | 0.061 | 0.055 | 0.050 |
| Day 3-movementSpeed | 0.069 | 0.065 | 0.059 | 0.050 |
| Day 1-restingDuration | 2.03 | 2.14 | 2.76 | 2.87 |
| Day 2-restingDuration | 1.40 | 1.66 | 1.94 | 2.59 |
| Day 3-restingDuration | 1.86 | 1.72 | 2.23 | 2.31 |
| Day 1-restingFrequency | 101 | 118 | 105 | 109 |
| Day 2-restingFrequency | 111 | 109 | 107 | 109 |
| Day 3-restingFrequency | 106 | 111 | 109 | 111 |

duration (*restingDuration*) on day 2 ($p < 0.05$). The heavy tag significantly decreased the average length of movement (*movementLength*) on day 3 ($p < 0.05$), the average maximum length of movement (*movementLengthMax*), the total distance travelled (*movementSum*), and the average movement speed (*movementSpeed*) on all 3 days ($p < 0.05$), and increased the resting duration (*restingDuration*) on day 2 and 3 ($p < 0.05$). Medians of these results are summarized in Table 4 and p-values are summarized in S1A Table.

Further analysis showed that these results strongly depended on temperature. Among the 54 data sets (6 movement properties x 3 days x 3 tag weights = 54) for each of the three temperatures, Mann-Whitney tests indicated that 36, 4, and 3 were significant at the low, intermediate, and high temperature, respectively (Table 5).

At the low temperature, the light tag significantly decreased the average length of movement (*movementLength*) on all 3 days ($p < 0.05$), the distance travelled (*movementSum*) and the average speed (*movementSpeed*) on day 2 and 3 ($p < 0.05$), and increased the number of stops (*restingFrequency*) on day 1 ($p < 0.01$). The medium tag significantly decreased the average length of movement (*movementLength*) on day 2 and 3 ($p < 0.05$), the maximum average length of movement (*movementLengthMax*), the distance travelled (*movementSum*), and the average speed (*movementSpeed*) on all 3 days ($p < 0.05$), and increased the duration of stops (*restingDuration*) on day 2 and 3 ($p < 0.05$). The heavy tag significantly decreased all distance movement properties (*movementLength*, *movementLengthMax*, and *movementSum*) and average speed (*movementSpeed*) on all 3 days ($p < 0.05$), and increased the duration of stops (*restingDuration*) on all 3 days ($p < 0.05$).

At the intermediate temperature, the light and medium tag did not significantly affect any movement property. The heavy tag decreased the distance travelled (*movementSum*) on day 3 ($p < 0.05$) and the average speed (*movementSpeed*) on all 3 days ($p < 0.05$).

At the high temperature, the light and medium tag did not significantly affect any movement property. The heavy tag significantly decreased the average speed (*movementSpeed*) on

**Table 5. Medians of movement properties of control crickets and crickets carrying light, medium, or heavy tags at three temperatures (low, intermediate, or high).** Gray highlighted values indicate significant Mann-Whitney tests ($p < 0.05$) for comparisons of movement properties of crickets with and without tags at particular temperatures.

| | low | | | | intermediate | | | | high | | | |
|---|---|---|---|---|---|---|---|---|---|---|---|---|
| | control | light | medium | heavy | control | light | medium | heavy | control | light | medium | heavy |
| Day 1-movementLength | 0.104 | 0.082 | 0.084 | 0.086 | 0.069 | 0.076 | 0.063 | 0.063 | 0.059 | 0.057 | 0.054 | 0.059 |
| Day 2-movementLength | 0.092 | l0.079 | 0.075 | 0.077 | 0.075 | 0.071 | 0.078 | 0.074 | 0.066 | 0.063 | 0.061 | 0.071 |
| Day 3-movementLength | 0.105 | 0.072 | 0.076 | l0.070 | 0.076 | 0.085 | 0.084 | 0.075 | 0.065 | 0.069 | 0.068 | 0.062 |
| Day 1-movementLengthMax | 0.515 | 0.443 | 0.327 | 0.341 | 0.345 | 0.304 | 0.283 | 0.247 | 0.241 | 0.311 | 0.230 | 0.224 |
| Day 2-movementLengthMax | 0.451 | 0.311 | 0.257 | 0.283 | 0.326 | 0.372 | 0.309 | 0.319 | 0.295 | 0.267 | 0.276 | 0.282 |
| Day 3-movementLengthMax | 0.528 | 0.396 | 0.291 | 0.334 | 0.365 | 0.449 | 0.354 | 0.303 | 0.291 | 0.295 | 0.310 | 0.282 |
| Day 1-movementSum | 22.05 | 17.07 | 13.47 | 10.69 | 16.32 | 12.32 | 14.43 | 9.19 | 7.97 | 10.36 | 7.39 | 6.93 |
| Day 2-movementSum | 22.56 | 16.08 | 11.33 | 11.28 | 19.70 | 18.17 | 15.90 | 14.91 | 17.56 | 15.39 | 16.64 | 11.60 |
| Day 3-movementSum | 21.87 | 16.35 | 12.15 | 11.73 | 18.04 | 22.91 | 17.84 | 12.98 | 11.40 | 16.47 | 10.73 | 10.05 |
| Day 1-movementSpeed | 0.067 | 0.056 | 0.048 | 0.051 | 0.066 | 0.061 | 0.064 | 0.055 | 0.061 | 0.054 | 0.056 | 0.047 |
| Day 2-movementSpeed | 0.063 | 0.050 | 0.048 | 0.042 | 0.067 | 0.064 | 0.060 | 0.053 | 0.066 | 0.063 | 0.061 | 0.053 |
| Day 3-movementSpeed | 0.065 | 0.048 | 0.044 | 0.040 | 0.069 | 0.073 | 0.065 | 0.055 | 0.069 | 0.070 | 0.066 | 0.053 |
| Day 1-restingDuration | 1.45 | 1.53 | 2.34 | 2.92 | 2.02 | 1.97 | 2.57 | 2.28 | 4.38 | 2.62 | 3.54 | 4.08 |
| Day 2-restingDuration | 1.25 | 1.50 | 2.18 | 2.60 | 1.49 | 1.75 | 1.81 | 1.66 | 1.65 | 1.72 | 1.71 | 2.65 |
| Day 3-restingDuration | 1.49 | 1.31 | 2.11 | 2.66 | 1.96 | 1.55 | 1.95 | 1.96 | 2.92 | 2.07 | 2.59 | 2.36 |
| Day 1-restingFrequency | 105 | 127 | 119 | 105 | 107 | 116 | 103 | 117 | 80 | 102 | 97 | 92 |
| Day 2-restingFrequency | 105 | 113 | 116 | 98 | 115 | 109 | 107 | 119 | 99 | 103 | 100 | 105 |
| Day 3-restingFrequency | 111 | 121 | 114 | 101 | 106 | 106 | 116 | 109 | 98 | 114 | 104 | 111 |

all 3 days ($p < 0.05$). The medians of these results are summarized in Table 5, all p-values are summarized in S1B Table.

We also tested the influence of temperature on the movement properties of the control crickets and for crickets within each tag weight group (e.g., for crickets with light tags at the three temperatures). Control crickets (i.e., without tags) travelled longer distances (*movementSum*) at the low temperature than at the medium temperature on day 1 ($p < 0.05$) and at the low temperature than at the high temperature on all 3 days ($p < 0.001$). Increase of distances was significant also in both single movements properties (*movementLengthMax*, *movementLength*) on all 3 days. Average speed (*movementSpeed*) did not differ among temperatures, but stops between single movements (*restingDuration*) were shorter at the intermediate temperature temperature on day 1 and 3 ($p < 0.05$) and at the high temperature on all 3 days ($p < 0.001$). The number of stops (*restingFrequency*) on day 1 was significantly lower at low temperature than at high temperature temperature ($p < 0.05$). In the case of crickets with tags, total distance travelled (*movementSum*) significantly differed between temperatures only for crickets with light tags. Unlike crickets without tags, crickets with light tags travelled shorter distances at the low temperature than at the intermediate temperature on day 3 ($p < 0.05$), but travelled greater distances at the high temperature than at low temperature on day 1 ($p < 0.05$). Stops (*restingDuration*) were shorter at the low temperature than at the high temperature on day 1 ($p < 0.05$). Speed of crickets (*movementSpeed*) with all tag weights was lower at the low temperature than at the intermediate or high temperature ($p < 0.01$). All medians are summarized in Table 5, and all p-values are summarized in S2 Table.

**3.1.1 Movement distance.** In the mixed linear model concerning movement distance, the p-value of the random effect (cricket identity) was $1.13 \times 10^{-13}$. According to ANOVA, movement distance was significantly affected by temperature (F = 20.05, $p < 0.001$), tag weight (F = 93.5, $p < 0.001$), animal weight (F = 49.1, $p < 0.001$), and by interactions between

**Table 6. Mixed linear model of the distance travelled (*movementSum*) by crickets over 3 days.**

|  | Estimate | Std. Error | df | t value | Pr(>|t|) |
|---|---|---|---|---|---|
| (Intercept) | 0.8481 | 2.4375 | 294.7989 | 0.3479 | 0.7281 |
| high temperature | -10.4746 | 3.5562 | 349.7777 | -2.9455 | 0.0034 |
| low temperature | 10.8655 | 3.2454 | 258.9370 | 3.3480 | 0.0009 |
| tag weight | -0.0077 | 0.0012 | 167.3106 | -6.3737 | 0.0000 |
| animal weight | 0.0217 | 0.0031 | 303.1101 | 7.1041 | 0.0000 |
| high temperature:tag weight | 0.0052 | 0.0017 | 168.0552 | 3.0456 | 0.0027 |
| low temperature:tag weight | -0.0063 | 0.0017 | 166.9246 | -3.6653 | 0.0003 |
| high temperature:animal weight | 0.0077 | 0.0044 | 360.8046 | 1.7496 | 0.0810 |
| low temperature:animal weight | -0.0109 | 0.0039 | 268.4809 | -2.7685 | 0.0060 |

temperature and tag weight (F = 14.05, $p < 0.001$) and between temperature and animal weight (F = 3.48, $p < 0.001$) (S3 Table). The interaction of temperature and tag weight was previously indicated by Mann-Whitney tests. According to the coefficients of the tag weight in intermediate temperature (-0.0077) and its interaction with low temperature (-0.0063), reduction in the distance travelled by a cricket in low temperature in 10 minutes corresponds to a 1.4m for every 100 mg of tag attached. All coefficients are summarised in Table 6 and the distribution of travelled distances of crickets is shown in Figs 4 and 5.

## 3.2 Effect of attachment duration

We analysed the distributions of slopes for all movement properties, and we compared the distributions of the slopes of the linear models were compared with the Wilcoxon test (the

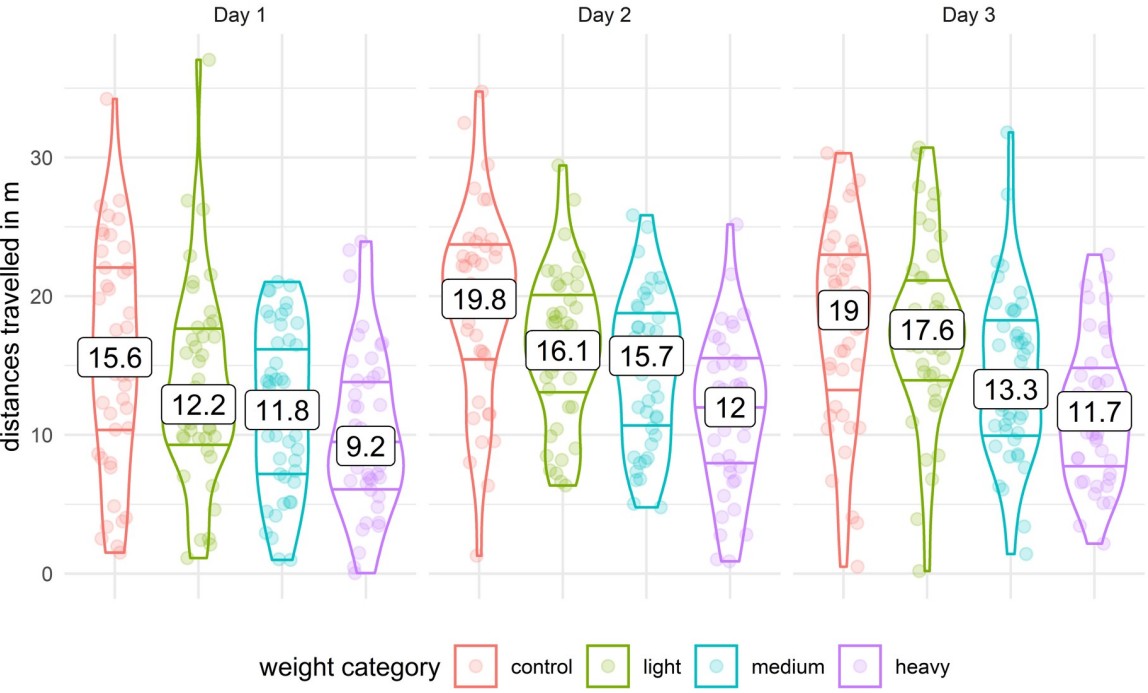

**Fig 4. Distribution of distances travelled (*movementSum*) by crickets carrying no tags (control) or carrying light, medium, or heavy tags over 3 consecutive days without regard to temperature.** The bottom horizontal lines represent the 1st quartiles, the boxes in the middle indicate the medians, and top horizontal lines represent the 3rd quartiles.

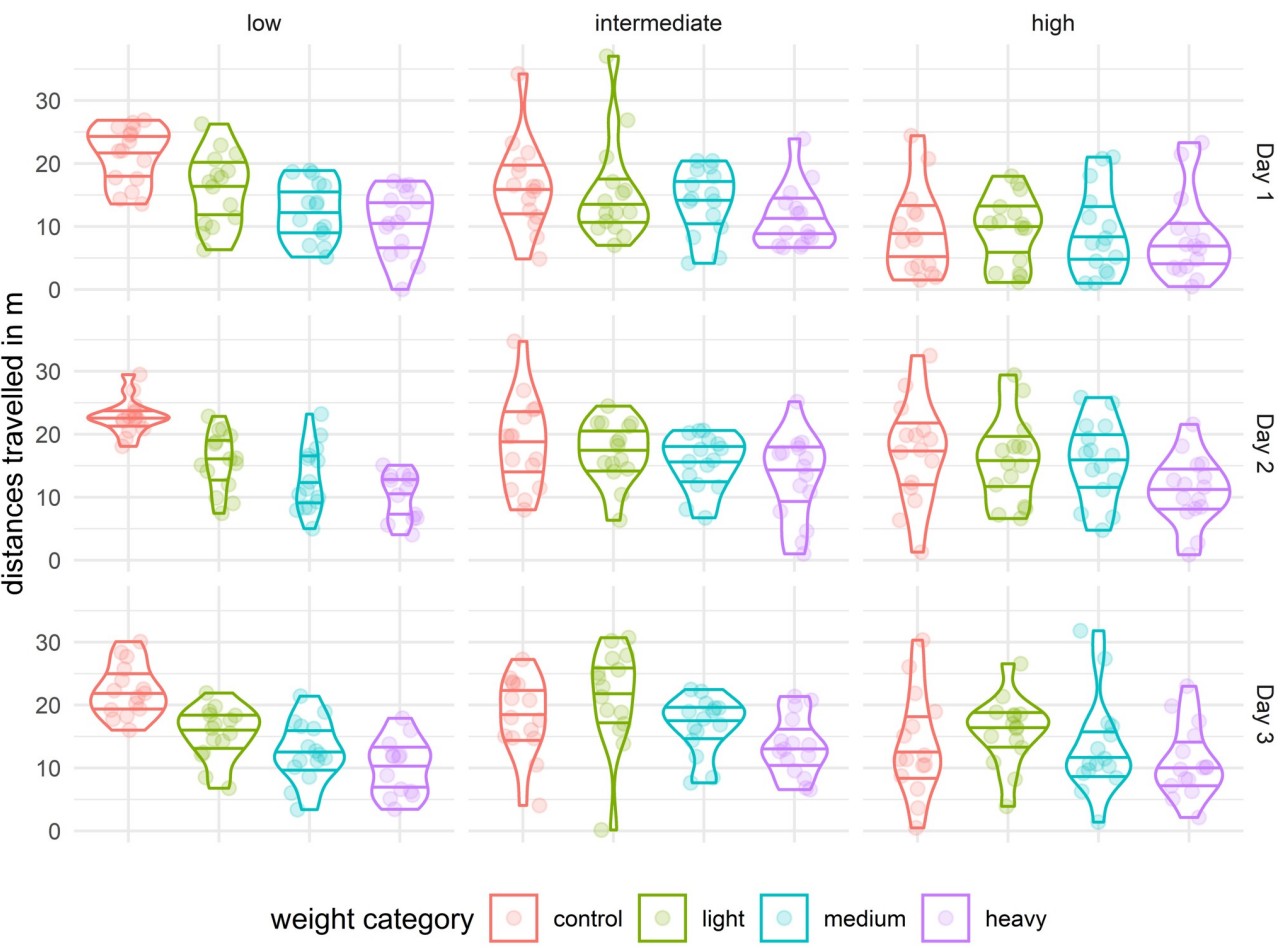

**Fig 5. Distributions of distances travelled (*movementSum*) of crickets on all 3 days as affected by low, intermediate, and high temperatures (indicated at the top).** The horizontal lines represent, from bottom to top, the 1st quartile, median, and 3rd quartile.

medians of the slopes and p-values are summarized in S4 Table). The only significant difference was in the average speed (*movementSpeed*; $p < 0.01$) between crickets in the control and heavy tag groups. We modelled the average speeds of crickets during the 3 days with simple linear models with day number as the only independent variable. The distributions of slopes of all models are shown in Fig 6. The median of the slopes was $<0$ only for crickets with heavy tags, i.e., the average speed of crickets with heavy tags but not of control crickets or crickets with light or intermediate tags decreased by 0–50% over the 3 days.

### 3.3 Body mass change

The mean weight (± SD) of individuals before recordings on the day 1 was 768 ± 117 mg, and the mean increased during the experiment (Fig 7). Across all tag weights, there was no significant shift between day 1 and day 2 ($p = 0.425$), but there was a significant increase between day 2 and day 3 ($p < 0.001$), with a 22-mg increase in the median. In each weight category, cricket weight was greater at the end than at the beginning of the experiment, although the weight of crickets with medium or heavy tags decreased between day 1 and day 2 and then substantially increased between day 2 and day 3 (see Fig 7).

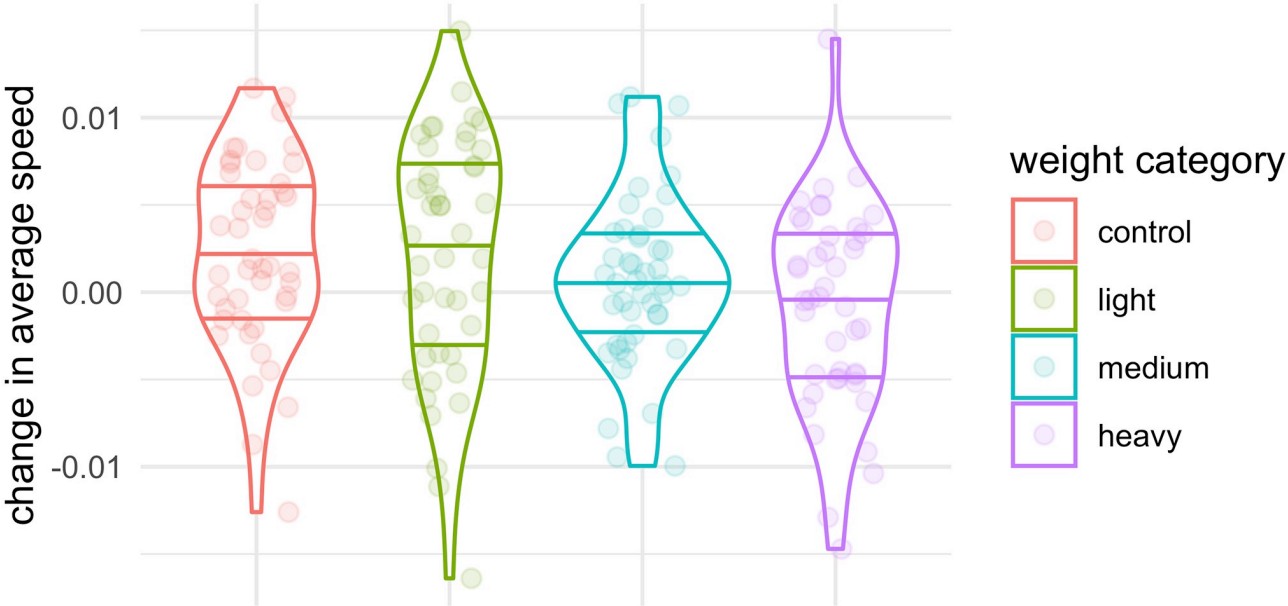

**Fig 6. Distributions of slopes of linear models of the change in average speed (*movementSpeed*) of crickets during 3 days across all three temperatures.** The horizontal lines represent, from bottom to top, the 1st quartile, the median, and the 3rd quartile.

## 4 Discussion

In our study, tag mass expressed as a percentage of cricket body mass ranged from 18 to 127%, and while all crickets were able to move with tags, whether light, medium, or heavy, we found

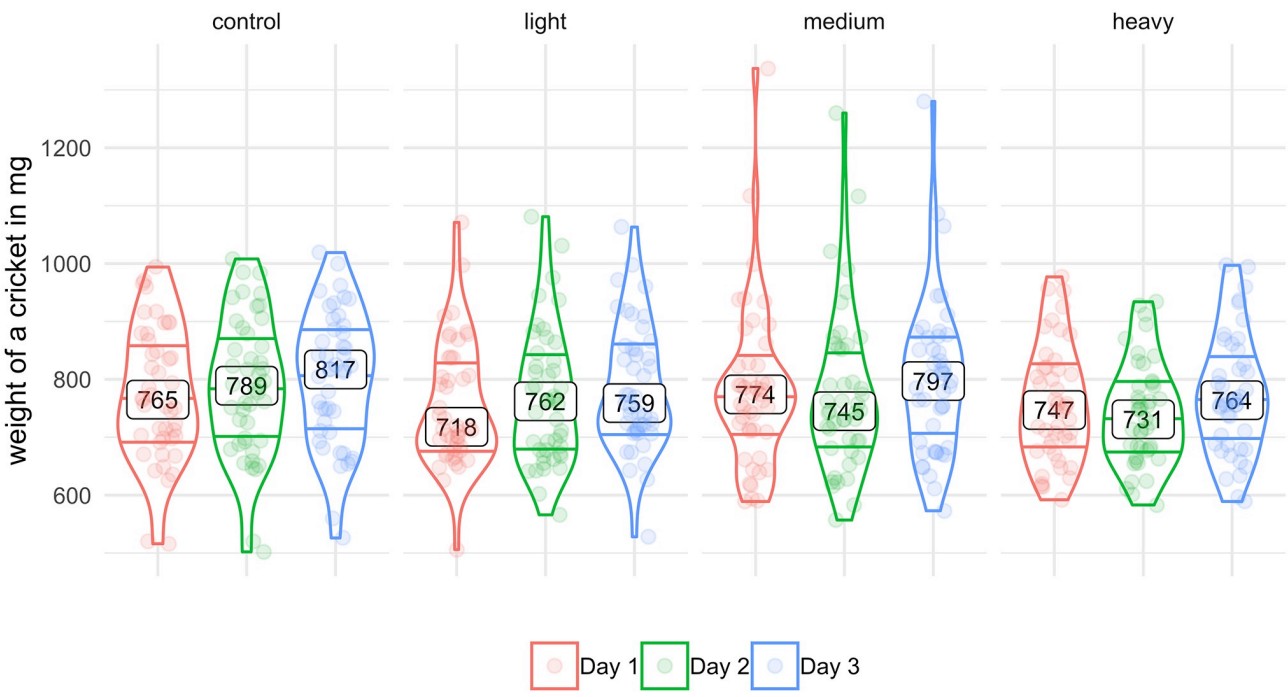

**Fig 7. The distributions of cricket weights on all 3 days without regard to temperature.** The bottom horizontal lines represent the 1st quartiles, the boxes in the middle indicate the medians, and top horizontal lines represent the 3rd quartiles.

that even light tags affected cricket movement. However, the effect was strongly dependent on temperature. Although results without regard to temperature revealed that even the light tag affected cricket movement, analysis of temperature indicated that the effects of the light tags were strongest at the low temperature. We didn't find significant effects on crickets movement properties with the light and medium tag at intermediate and high temperatures.

Our results indicate that, while tags did not affect the movement based on a qualitative (visual) assessment, a quantitative assessment revealed that tags can reduce cricket movement, as was previously noted by Boiteau et al. [37]. An insufficient evaluation of the effects of tags on animal movement can cause biased estimates of movement abilities and can affect the application of the results in ecosystem science, pest control, or conservation [3]. In some cases, however, a qualitative assessment of the effects of the added weight is sufficient. Kennedy et al. [7], for example, used radio telemetry to locate the nests of the invasive Asian hornet *Vespa velutina* Lepeletier, 1836; the authors used a preliminary test to simply determine whether the hornets were able to fly with transmitters, because locating the nests rather than understanding movement was the objective. The use of a heavy tag (e.g., equal to the weight of the insect) is reasonable if the research is not directly interested in how the tag affects movement properties.

Regardless of temperature, the attachment of the light tag (ca. 27% of cricket weight) decreased distance travelled by 19%, speed by 10% and increased number of stops by 17%. Previous studies have shown varying degrees of influence of light tags, i.e., tags representing up to 30% of animal weight. Lorch et al. [8] found no effect on speed, distance, and turning behaviour of a tag whose weight was ca. 15–25% of that of the *Anabrus simplex*. A transmitter load of 8–23% also did not affect the distance travelled and the numbers of flights of the stag beetle *Lucanus cervus* (Linnaeus, 1758) [55]. Similarly, the flight speed and some other performance indices of the lepidopteran *Anartia fatima* Godart were not affected by a tag weight that was 15% of the body mass [56]. On the other hand, tags weighing 2.2 and 3.9% of average body mass decreased the speed of movement of *Leptinotarsa decemlineata* (Say) and *Conotrachelus nenuphar* (Herbst) by 8 and 36% respectively [37]. In another example, the flight behaviour of the hemipteran *Ricania* sp. was negatively affected by a tag that weighed ca. 10% of the body weight [57].

In our experiment, crickets with medium tags (ca. 72% of cricket weight) and heavy tags (ca. 102% of cricket weight) walked up to 17% and 27% slower, respectively, than crickets without tags; the tags also reduced the distanced travelled (by 30% and 41%, respectively), their maximum single movement (by 28% and 29%, respectively), and increased the duration of their stops (by 38% and 85%, respectively). Increased resting periods were previously observed for bumblebees *Bombus hortorum* L. with attachments representing of 44–66% of bumblebees body mass [5]. Our results not surprisingly indicate that the heavier the tag, the greater the effect. We therefore confirm that researchers should reduce the tag/body mass ratio as much as possible, but that the effects of the tag/body mass ratio can be species-specific [37].

As noted at the beginning of the Discussion, the effect of tag mass on cricket movement strongly depended on temperature. Previous research suggested that the effect of tag attachment is dependent on environment and stress in the case of birds [26], and arthropods [3]. In the case of *G. locorojo*, we found that the low temperature (19.5˚C) negatively affected the majority of movement properties regardless of tag weight (see Fig 4). At the low temperature, the light tag reduced the total distance travelled by 29%, movement speed by 20%, and increased number of cricket stops by 21%. These effects of low temperature were even greater with medium and heavy tags, with the exception of stops, where the number did not increase, but the duration. Detailed analysis of the distance travelled in low temperature showed that cricket movement in 10 minutes would be reduced by 1.4 m for every 100 mg of tag weight. With increasing temperature, the magnitude of change decrease. At the intermediate and high

temperatures, significant effects on of the movement properties was confirmed only with heavy tag. The heavy tag slowed the crickets in the intermediate and high temperatures, but distance travelled was decreased only in a single day at the intermediate temperature. Surprisingly, we found that crickets with light tags at intermediate and high temperatures tended to walk faster and travel farther than crickets without tags (S1B Table). This surprising effect has also been previously documented: Boiteau et al. [37] found that the mean speed of movement of *Diabrotica virgifera virgifera* (LeConte) increased after tag attachment (ca. 5% of the body weight). These results indicate that environmental conditions alter with tag effects [3] and suggest that in some cases (e.g., tracking daytime activity on a site) it is possible to use relatively heavy tags for experiments if the possible effects are accounted for. Conversely, research results may be biased if field conditions change significantly during survey of insects with tags. The effect of temperature may also differ between insects that can or cannot fly, because the former can regulate their body temperature by flying or shivering [58].

It is well-known that temperature can strongly alter insect movement [27, 28]. In the current study, the crickets without tags generally travelled longer distances at the low temperature than at higher temperatures; their speed did not vary but they stopped for shorter times at lower than at higher temperatures. Perhaps the crickets at the low temperature were seeking a more suitable place for thermoregulation [59]. Interestingly, unlike the crickets without tags, crickets with medium and heavy tags did not show any differences in the distance travelled between temperatures. The speed of crickets with any attachment, however, was decreased more at the low temperature than at the higher temperatures. The temperatures tested here were not extreme. Although we know little about *G. locorojo* biology [38], the temperatures in previous experiments with this species ranged from ca. 23 to 27°C [40–42], but the species has been reared at temperatures up to 32°C [38]. It is possible that at higher temperatures than tested in the current study, the mobility and associated effect of tag attachment may change. Other studies have found that high temperatures reduce the running speed and jumping distance of *Acheta domesticus* L. (Orthoptera: Gryllidae) [27] and can cause orthopterans to move to cooler locations [60, 61]. Based on our results, we suggest that the effect of additional weight attachment on movement properties may differ in combination with altered behaviour under suboptimal environmental conditions. Additional research is needed, however, on the combined effects of temperature and tags on the movement of other insect taxa.

Effects of tag weight on *G. locorojo* movement varied from day to day in the current study. Our results indicate that the duration of tag attachment is not related with the influence of the tag weight on *G. locorojo* behaviour, except that the average movement speed of crickets with heavy tags decreased with attachment duration. This may be related to the insect's metabolism, because the extra weight may increase the insect's energetic cost, as suggested by Kissling et al. [1]. Lee et al. [16] found that tag attachment with various glues (the extra weight was ca. 2.3% of the body weight) did not affect *Halyomorpha halys* survivorship or distance travelled in 7 days of attachment.

For *G. locorojo* crickets in the current study, body mass with or without a light tag slightly increased during the 3 days of the experiment. Although the body mass of crickets with medium or heavy tags (but not with light tags) declined between the start of the experiment and day 2, the body mass of crickets with medium and heavy tags increased between on day 2 and day 3, such that the crickets were heavier on day 3 than at the beginning of the experiment. Perhaps the crickets with medium and heavy tags compensated for the loss of body mass on day 2 by eating more to regain weight. This would be consistent with the results of previous research with tagged *Deinacrida heteracantha*, whose body weight increased during a 10-day study [62]; in that study, however, the ratio of tag mass to body mass was not reported. In our study, the mean weight of crickets measured before the first recordings with or without tags

was greater at the end than at the beginning of the whole experiment. Perhaps this is a natural process involving the maturation of the ovaries and the increase in the storage of the fat needed to supply the energy for reproduction.

We recognize that the results of our laboratory study, at least in their details, may be species-specific and may not correspond to the situation in the field. In particular, individuals were fed *ad libitum* in our experiment, and the effects of tag weight could differ when animals have limited access to food, because food availability can affect dispersal ability [63] and body mass [64]. Our experimental arena lacked heterogeneity and therefore lacked shelters, which under natural conditions can be used for thermoregulation [59]. In addition, our experiment did consider tag weight but did not consider other factors associated with tags that could affect insect mortality or behaviour (see [3]). Boiteau et al. [18], for example, reported high mortality of *D. virgifera virgifera* due to cyanoacrylate adhesive application. The opposite effect was observed by Niemelä et al. [65], who described that RFID tag attachment with cyanoacrylate adhesive did not affect survival of *Gryllus campestris* Linnaeus, 1758 during 20 days of field observation. In the current study, we did not observe any mortality of *G. locorojo* following adhesive application. In addition, our tags lacked antenna wire, but such wires can alter the movement or other aspects of behaviour in the field. If tagged insects burrow, the antenna may bend or break and thereby reduce the detection range [66]. Finally, we examined only flightless species, while the effect of tag attachment has its specifics to flying species. Not only the weight itself, but also, for example, the allocation of the central mass of the body after attachment, can affect the movement of flying insects. [67]. Additionally, it has also been found in small birds, that the increased drag caused by the tag had a similar impact as the tag weight [25]. Despite its limitations, our experiment has demonstrated that the effects of tags and temperature on arthropod movement should not be ignored in experiments that use tags.

Unlike previous studies using tracking software [8, 16, 18, 30, 37], we used an algorithm with identity tracking [48] that allows researchers to retrieve trajectories of many individuals from a single video recording; this increases the rate of data collection and analysis, and allows researchers to obtain trajectories for multiple individuals under identical conditions. We assume that the possibility of automatic video analysis will increase with the continued development of visual tracking algorithms. Video analysis is improving not only in terms of precision and speed but also in terms of simplicity and affordability, as demonstrated by *AnimApp* for Android devices [68] or other methods, e.g., 3D trajectory segmentation for flying insects [69]. We also suggest that with the increased use of various technologies in ecological surveys, it will become possible to directly estimate the effects of transmitters in the field; this could be done, for example, with fluorescence detection using UAVs (unmanned aerial vehicles) [70] in combination with image/video analysis.

The use of UAVs and other ongoing technological advances are creating further possibilities for tracking the movement of insects and will likely increase the use of tracking tags in insect ecology [11, 20, 71–73]. Furthermore, there are non-tracking cases in which insects are used to carry devices [20, 21]. We agree with other researchers [3, 19, 37] that the effects of tags are likely to be species-specific, i.e., there are no general thresholds or rules for tag/body mass ratios that can be applied to all species. We also agree that the effect could differ with environmental conditions [3, 26]. We recommend that researchers consider or investigate the possible effects of tags before conducting any experiment with tags in order to avoid obtaining biased results.

## Supporting information

**S1 Table. Tests for the effect of tag weight on the movement properties of crickets.** P-values of Mann-Whitney tests of movement properties of control crickets and crickets carrying light,

medium, or heavy tags with and without regard to temperature.
(PDF)

**S2 Table. Effect of temperature on movement properties for crickets in separate tag weight categories and for control crickets.** P-values of Mann-Whitney tests for the effect of temperature on movement properties of crickets separately for each weight category (light, medium, heavy) and for control crickets.
(PDF)

**S3 Table. Effects of the indicated variables on the distance travelled *movementSum* by crickets over 3 days.**
(PDF)

**S4 Table. Change in movement properties during 3 consecutive days.** Medians and Wilcoxon test of differences in distributions of slopes of linear models that describe change in movement properties as affected by tag weight.
(PDF)

## Author Contributions

**Conceptualization:** Oto Kaláb, David Musiolek, Petr Hurtik, Petr Kočárek.

**Data curation:** Petr Hurtik.

**Formal analysis:** Pavel Rusnok, Petr Hurtik.

**Funding acquisition:** Oto Kaláb, Petr Kočárek.

**Investigation:** Oto Kaláb, David Musiolek, Petr Hurtik, Petr Kočárek.

**Methodology:** Oto Kaláb, David Musiolek, Petr Hurtik, Martin Tomis.

**Project administration:** Oto Kaláb, Petr Kočárek.

**Resources:** David Musiolek, Petr Hurtik, Martin Tomis.

**Software:** Petr Hurtik.

**Supervision:** Oto Kaláb, Petr Kočárek.

**Visualization:** Pavel Rusnok.

**Writing – original draft:** Oto Kaláb, Pavel Rusnok, Petr Hurtik.

**Writing – review & editing:** Oto Kaláb, David Musiolek, Pavel Rusnok, Petr Hurtik, Petr Kočárek.

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
