## [Decision Letter · Decision Letter 0]

24 Mar 2021

PONE-D-21-05023

Estimating the effect of tracking tag weight on insect movement using video analysis: a case study with an orthopteran

PLOS ONE

Dear Dr. Kaláb,

Thank you for submitting your manuscript to PLOS ONE. After careful consideration, we feel that it has merit but does not fully meet PLOS ONE’s publication criteria as it currently stands. Therefore, we invite you to submit a revised version of the manuscript that addresses the points raised during the review process.

Both reviewers had very similar comments. I also read the manuscript, and we all think that the paper would be a useful contribution to radio tracking insects, but the manuscript needs to be made much clearer in organization and the statistics need to be refined. If you think you can revise the manuscript in accordance with the reviewers' suggestions, then I would be pleased to see a revised version with a detailed reply. However the revised manuscript may go to the same or different reviewers, and there is no guarantee that it will be accepted.

We look forward to receiving your revised manuscript.

Kind regards,

Robert B. Srygley

Academic Editor

PLOS ONE

Journal Requirements:

'We acknowledge support from University of Ostrava students grant (SGS14/PrF/2018) 471

and the Orthopterist's Society by The Theodore J. Cohn Research Fund.'

'The funders had no role in study design, data collection and analysis, decision to publish, or preparation of the manuscript.'

Reviewers' comments:

Reviewer's Responses to Questions

**Comments to the Author**

1. Is the manuscript technically sound, and do the data support the conclusions?

Reviewer #1: Partly

Reviewer #2: Partly

2. Has the statistical analysis been performed appropriately and rigorously? 

Reviewer #1: No

Reviewer #2: I Don't Know

3. Have the authors made all data underlying the findings in their manuscript fully available?

Reviewer #1: Yes

Reviewer #2: Yes

4. Is the manuscript presented in an intelligible fashion and written in standard English?

Reviewer #1: Yes

Reviewer #2: Yes

5. Review Comments to the Author

Reviewer #1: This manuscript describes a study using commonly available crickets to study the effects of added weight from tags attached to track insect movement. They also look at whether temperature increases modify these effects. They do a reasonable job of placing this work in the context of existing research. The researchers present a novel way of building tags of the same size with different weights and use a video analysis technique that has properties desirable for small groups of moving insects. They state that the methods presented could and should be used by anyone using any kind of tag to assess the effects of the added weight on insect behavior.

One major flaw that I see with this paper is that temperature is presented as an independent variable, but it is confounded by time in the arena. Temperature was always increased during the course of an experimental run rather than having animals randomly assigned to a temperature treatment or temperature regime order. So you cannot say whether changes in movement were due to temperature or just due to being in the arena longer on a given day. There are several possible solution to this problem. The simplest way to deal with this is to eliminate temperature as a consideration. Another is to treat temperature as a covariate and refocus the discussion on temperature by treatment interaction terms. Finally, you could do an analysis similar to the slope analysis you used to look at the effects of number of days tested. In the Mann-Whitney tests, done separately by temperature category, and in the mixed linear models (where temperature appears to have been entered as categorical as indicated by Df = 2). You will also need to reframe all descriptions of temperature effects in terms of “increasing temperature.”

Your description of the video analysis technique is too detailed for a paper on movement effects given that it is published elsewhere. You should refer readers to the other paper for details and mention why you feel your methods are better than other available software. You should probably also mention recently published tracking software such as the recently published TRex (https://www.biorxiv.org/content/10.1101/2020.10.14.338996v3.full.pdf).

I think it is worth mentioning that for the insect you focus on, these tag weights are relatively heavy. You should also mention the average weight of female crickets when describing the study species. Readers can see weights in some figures much later in the paper, but not when they are evaluating tag weight in the methods.

You do an awful lot of tests of the same hypotheses across days, and temperatures. Some of these tests would be significant just by chance. Not only does it raise a statistical issue, but it makes your results overly complicated. Eliminating the temperature effects tests table will help make the results more approachable, but you should probably also be using significance levels on tests that are adjusted for the number of tests.

Minor comments by line number:

Line 9: light weight to lightweight

17: remove “(less than 1 week)”

24: I would use post hoc rather than expost

23: affection to effects

28: weighted to weight

162: propertie to property

171: repeated measuring to repeatedly measuring

181: heavy needs to be capitalized

195: Add “movement” before property

264: Remove animals

309: replace but with a comma

386-389: Remove stuff about High frequency radiation. It seems irrelevant.

428: Solar tags have been in use for birds for a number of years, so they are not just being considered.

In the discussion you mention that you think the effects of tag weight are species specific in at least two places. Consider combining them.

Reviewer #2: It seems like it has potential to be a good paper, but right now it is not written in a clear or effective way. Thee authors need to clearly define why the study was conducted and what benefit this study is providing the field. I'm often left asking "why?" and had a hard time following the flow of ideas.

I think the stats are too complicated because they added temperature as a treatment in addition to transmitter weight. I am really not sure why this was done and think easier stats would be more clear for readers.

6. PLOS authors have the option to publish the peer review history of their article (what does this mean?). If published, this will include your full peer review and any attached files.

Reviewer #1: No

Reviewer #2: No

---

## [Author Response · Author response to Decision Letter 0]

12 May 2021

Reviewer #1: 

One major flaw that I see with this paper is that temperature is presented as an independent variable, but it is confounded by time in the arena. Temperature was always increased during the course of an experimental run rather than having animals randomly assigned to a temperature treatment or temperature regime order. So you cannot say whether changes in movement were due to temperature or just due to being in the arena longer on a given day. There are several possible solution to this problem. The simplest way to deal with this is to eliminate temperature as a consideration. Another is to treat temperature as a covariate and refocus the discussion on temperature by treatment interaction terms. Finally, you could do an analysis similar to the slope analysis you used to look at the effects of number of days tested. In the Mann-Whitney tests, done separately by temperature category, and in the mixed linear models (where temperature appears to have been entered as categorical as indicated by Df = 2). You will also need to reframe all descriptions of temperature effects in terms of “increasing temperature.”

In response to the reviewer’s concern, we have clarified the text. We treated temperature as an independent variable because in our experiment every run was recorded at a particular constant temperature, i.e., there wasn’t any increase during a single experimental run. In the beginning of the whole experiment, 180 crickets were randomly assigned to 9 independent groups (each with 20 crickets); 3 groups were separately tested at a low, intermediate, and high temperature. Each group was in the arena for 15 +10 min (habituation + data acquisition) at the particular constant temperature. In other words, the time in the arena was the same for all temperatures.

Your description of the video analysis technique is too detailed for a paper on movement effects given that it is published elsewhere. You should refer readers to the other paper for details and mention why you feel your methods are better than other available software. You should probably also mention recently published tracking software such as the recently published TRex (TRex, a fast multi-animal tracking system with markerless identification, and 2D estimation of posture and visual fields).

Based on the reviewer’s concern, we have significantly reduced the description of our software. We have also added an overview of the TRex software.

I think it is worth mentioning that for the insect you focus on, these tag weights are relatively heavy. You should also mention the average weight of female crickets when describing the study species. Readers can see weights in some figures much later in the paper, but not when they are evaluating tag weight in the methods.

Accepted. We added information about average cricket weights and the range of mass of used tags expressed as a percentage of cricket body mass to the Model species subsection.

You do an awful lot of tests of the same hypotheses across days, and temperatures. Some of these tests would be significant just by chance. Not only does it raise a statistical issue, but it makes your results overly complicated. Eliminating the temperature effects tests table will help make the results more approachable, but you should probably also be using significance levels on tests that are adjusted for the number of tests.

Thank you for your valuable comment. In the revised manuscript, we adjusted significance levels according to Benjamini & Hochberg 1995 - Controlling the False Discovery Rate: A Practical and Powerful Approach to Multiple Testing. We also replaced the results tables with tables of medians, which provide a clearer picture of the calculated values of movement properties. Tables with the results (p-values) were moved to the supplementary material.

Minor comments by line number:

Line 9: light weight to lightweight

changed

17: remove “(less than 1 week)”

removed

24: I would use post hoc rather than expost

changed

23: affection to effects

changed

28: weighted to weight

fixed

162: propertie to property

fixed

171: repeated measuring to repeatedly measuring

fixed

181: heavy needs to be capitalized

changed

195: Add “movement” before property

added

264: Remove animals

removed

309: replace but with a comma

replaced

386-389: Remove stuff about High frequency radiation. It seems irrelevant.

removed

428: Solar tags have been in use for birds for a number of years, so they are not just being considered.

This passage was discarded in response to the comments of the second reviewer. 

In the discussion you mention that you think the effects of tag weight are species specific in at least two places. Consider combining them.

We merged these paragraphs, as recommended.

Reviewer #2:

It seems like it has potential to be a good paper, but right now it is not written in a clear or effective way. The authors need to clearly define why the study was conducted and what benefit this study is providing the field. I'm often left asking "why?" and had a hard time following the flow of ideas.

In the revised manuscript, we have attempted to clarify why the study was conducted and how it benefits the field.  

I think the stats are too complicated because they added temperature as a treatment in addition to transmitter weight. I am really not sure why this was done and think easier stats would be more clear for readers.

We agree that the description of the experimental design was confusing in the previous version. We have attempted to correct this in the revised manuscript (discussed in greater detail in responses below).  

Attachment

Summary:

This is a great approach!!! This method will speed up the process of determining if insects are impacted by the attachment of a tracking tag. There are really great aspects of this research! However, it would benefit from a rewrite keeping in mind WHY the study was conducted. It is currently vague and disjointed. I would like more justification for decisions made and a more descriptive description of the results. The part about temperature is not clearly described – I am left wondering why temperature was included as a treatment and not just as “we maintained temperature between x and y degrees Celsius because those are typical conditions”

In the revised manuscript, we have clarified why the study was conducted and why temperature was considered as a variable. 

Introduction

Overall the introduction could benefit from more specific information. Currently, it reads like a brief overview and is not as compelling as it would be with specific examples and information. Why is this important? What makes this study unique? What methodological problem does this solve?

We have revised the Introduction in response to the reviewer’s comments. The Introduction now clearly describes the problem of tag effects and why is it important to estimate these effects in future studies. We have moved two paragraphs about video analysis from the Discussion to the Introduction as suggested in later comments, and we have added what differences make our approach.

1st paragraph (lines 2-20) – Give an estimate of what you mean by light and heavy radio telemetry tags. How heavy are heavy tags and how long do their batteries last? How light are light tags and how long do their batteries last? What do you mean by smaller animals? This paragraph overall feels vague and would benefit from more specific examples from the literature.

In response to the reviewer’s concern, we have rewritten this paragraph. We have reduced the overview information and focused on explaining the problem of tag weights. We have added specific information about radio telemetry tags (weights, battery life) with specific examples of studies that used these tags.

2nd paragraph (lines 21-36) – This paragraph changes topic from how studies account for tag attachment to the impact of temperature to tags outside of radio telemetry and then finished with vertebrates. The ideas are good! But the flow is confusing. Could benefit from splitting into 2 or several paragraphs. Remember the 1st sentence of the paragraph should act as a topic sentence for the entire paragraph. – Maybe one paragraph on the lack of substantial data on tag effects on animals (insects and vertebrates) and how it could be for any type of tag, not just radio telemetry. Then a paragraph on the influence of temperature of insect movement and how it could relate to tag attachment.

As recommended by the reviewer, we have divided this paragraph into two paragraphs. We added information about 1) how tags could affect properties other than movement, 2) whether the effects are species-specific, and 3) why it is important to consider the effects. The information about the possible effects of temperature has been moved to a separate (2nd) paragraph.

3rd paragraph (lines 37-46) – Add a little more justification for why you think different weights and different temperatures will influence movement. Also, what is the justification of the effect changing over time? Justification of affecting cricket body mass?

In rewriting the previous paragraphs, we attempted to justify the questions studied. As suggested, we also briefly summarize this in the last paragraph.

Methods

 1. Model species: Be more specific with details. What is their range in size rather than just saying “sufficient size for the experiment”. What was provided as food? How many crickets were in each container? Why did you use 2nd  generation?         

We have provided additional information on the model species and its rearing. We have clarified the statement “sufficient size for the experiment”. Size in this context was related to the scale of tag mass expressed as a percentage of cricket body mass.

 2. Dummy tags: how many tags did you use at each weight? How does the weight compare to the cricket size? How were tags attached to crickets? Rather than having the info about the comparative tags from ATS and lotek (also should include Holohil) in supplemental, put this information in the manuscript. This is important info, especially if someone is planning to use your information to design a tracking study.

We added information about a number of constructed tags of each type. We moved the comparison of weight tags and crickets and attachment details from the Experimental design subsection to the Dummy tags subsection. We moved the table with telemetry tags from supplementary information to the manuscript with the addition of three Holohil-appropriate tag models.      

 3. Arena: line 78 ligth should be light. How many arenas did you have?

We had only one arena. This has been clarified in the revised manuscript.

4. Experimental design: first paragraph is confusing. Remember the 1st sentence should be a topic for the entire paragraph. What did the marking mean? Weighing the dummy tag and that superglue was used to attached to the cricket maybe should be included in the dummy tag section? Why did you only record their movement for 10 minutes? Why did you use the same crickets at the same temperature each day? Interesting that you chose to test tags that were heavier than the insects – good! What were you using to measure the impact of tag attachment on behavior? – later text makes it sound like you watched more than one cricket in the arena at a time. That is not clear here. Why did you do that?

First sentence about which crickets were used are composed in the Model species subsection. This results in the more consistent first paragraph. We clarified in text, that the markings are needed to unambiguously distinguish between individual crickets. We moved the comparison of weight tags and crickets and attachment details from Experimental design subsection to Dummy tags subsection as suggested in previous comment. We have explained why we recorded 10-min videos. The reason for recording the same individuals in the three consecutive days has been also explained in this section. The recording of multiple crickets at the same time speed-up the accomplishment of data and finishing the whole experiment. Using only one individual for each recording would require much more repetitions and video recordings, which would result in inefficiency of the whole method. The above mentioned rationale should be apparent from the whole paper.

5. Transforming visual data into numerical data: trajectory refers to a projectile. Haven’t provided enough background information for this section to make sense. What was recorded? What were the positions? What behaviors did crickets perform during their 10 minute observation period? What are the “unique locations”? should include camera information in the experimental design section. I think the second paragraph could be said more briefly and combine it with the information about your team’s software. Why didn’t you use etho vision?

Based on your valuable comment, we have completely rewritten the section Transforming visual data into numerical data and added a demonstration movie (Experimental design section) that helps explain the process. But we still use the term trajectory because it is a standard term in image processing and object tracking. The reason why we did not use Noldus EthoVision software is now described in detail in the response to one of your following comments. In brief, we didn’t use Noldus EthoVision software because it does not enable identity tracking, such that obtained results would not satisfy the research objectives.

 6. Statistical analysis: I would include the variables that were measured earlier in the section, not just in the statistical analysis section. Would include a more descriptive definition in table 2 for each property name.     

We moved the table of variables to the Experimental design section. The movement properties weren’t measured but were calculated from spatiotemporal data retrieved from analysis of video records. We acknowledge, however, that this part doesn’t fit well in the Statistical analysis section but fits better between Transforming visual data into numerical data and Statistical analysis. However, to avoid further fragmentation of the text, we renamed the Statistical analysis section as Data processing and statistical analysis, which better describes this process.

Results

Overall: Instead of trajectory I’d say step or movement length. Trajectory refers to a projectile and makes me think of the overall distance rather than the distance of one step of movement. Give specific examples and descriptions of your results. Mean or range in the measures. Rather than just saying significantly different, include the values. Crickets in the light, medium, and heavy categories moved xx, yy, and zz, cm, respectively. This will show if they are different because the ones in the heavy category didn’t move at all or if they moved, but just less than the other categories. Figures should indicate where statistical significance is present. The text should be able to stand alone without the figures and the figures should be able to stand alone without the text. Then when you have them together the reader should understand the whole picture. This needs to be done for every section of the results. Currently, the results section is very vague.

“Trajectory” is a standard term used in image processing and object tracking and also in movement ecology (e.g., GPS or telemetry data) see Edelhoff et al. 2016 Path segmentation for beginners: an overview of current methods for detecting changes in animal movement patterns https://doi.org/10.1186/s40462-016-0086-5. In the revised manuscript, we were careful to distinguish among the terms that refer to trajectory, total distance travelled, and single movement distance. We tried to include the values of median shifts in the text, but doing so made text too complicated. We therefore decided to replace the tables that had p-values (Tables 4 and 5 in the original manuscript) with tables with medians with significant results highlighted (the former tables are now available as supplementary S1). We think that this solution keeps the text concise and improves continuity and clarity. The plots were primarily meant to inform the reader about the distribution of the data (each dot refers to one cricket) rather than to show significant results. The visualized information about values and significance are now presented in Tables 4 and 5 in the revised manuscript. We then decided to keep the figures unchanged and as simple as possible. Fig. 4 and 5 represent only one, unlike the tables, which show all of the movement properties.

Discussion

Why didn’t you use etho vision – it shows to work well for studying insect movement with video? Did you have more than one cricket in the arena at one time? Why? Wouldn’t you think that the presence of other crickets could influence their behavior? What about when the crickets were housed over night between their 10 minute observation periods? I would think you would want them to be held in isolation to control their interactions and experiences. What is your justification for your choices? Why is it better to conduct trials with more than one individual at one time?

We attempted to use Noldus EthoVision software. Because it is quite expensive, we communicated with their sales department, and after a long negotiation, they provided us with a version of their software that met some but not all of our requirements. Most importantly, the software that they provided does not enable identity tracking and therefore could not be used to satisfy the goals of our study. Based on your comment, we have emphasized this in the revised text. As stated in the paper, we tracked 20 individuals at one time. 

It is better to conduct trials with more than one individual at one time because crickets interact with each other, i.e., they commonly occur in aggregations in the field (e.g., Villarreal et al. 2018, DOI: 10.1111/eth.12816. It follows that cricket movement is likely to be affected by the presence of other crickets. In addition, individual recordings would increase the variability in overall conditions: air temperature, humidity, barometric pressure, olfactory stimuli, part of the day, or other unknown factors that could influence the movement of individual crickets. Multiple-cricket recording provides homogenous environmental conditions and excludes a possible time-effect which may provide more consistent results. We assume that separating crickets in the inter-recording periods would have only a minor effect on the studied movement features.

The first paragraph feels very out of place. I would probably put that information in the introduction?

We moved this paragraph to the Introduction.

Video tracking software: this information should be included as justification for the methods you selected not in the discussion

We moved this paragraph to the Materials and methods.

Effects of tagging on the movement of Gryllus locorojo:

This should be the start of your discussion. Everything prior to this should find a new home in the intro/methods sections.

We revised the Discussion. As suggested by the reviewer, these paragraphs have been moved to the Introduction or Materials and methods.

It feels like a confusing story when temperature is included. Why did you use this as a variable? Is it a biologically relevant question? Why? How would this impact studies in the field?

We clarified this in the Introduction of the revised paper. Researchers previously documented that the effect of tags on bird movement depends on the environmental context (Snijdets et al. 2017). That the effect of tags changes with environmental conditions was also discussed in a recent review (Batsleer et al., 2020). Because insect movement is greatly affected by temperature, we determined whether the effect of tag attachment changes with temperature. We explain the reasons for studying temperature in the revised manuscript.

Discussion needs to be filtered down to tell a clear and concise story. Currently it feels like it is bouncing between ideas and I’m having a hard time trying to identify the point you are trying to make.

We have revised the Discussion in order to make it more concise and clear. 

Very cool to see even light tag effect distance traveled – I would put more emphasis on what happened because of the tags and not what happened because of temperature. If someone is conducting a field study, the insects will be under field environmental conditions… so I’m missing the point of the temperature variable.

In the revised paper, we have explained why we used temperature as a variable. In the revision, we consider the effects of tag weight with and without regard to temperature. We found that temperature alters the effects of tag weight on cricket movement. For example, crickets with light and medium tags were unaffected at the intermediate and high temperature but were significantly affected at the low temperature. We have added to the Discussion a consideration of the significance of this for field studies. Overall, our data demonstrate that the negative effects of tracking tags on cricket movement depend on temperature, i.e., temperature should be taken into account.  

What is the conclusion? Could tags of these weights be used for crickets? Why do you think that?

Our conclusion is that the safe use of tags (without biasing the resulting data) depends on the species, environment, and aim of the specific study. We therefore recommend investigating possible tag effects for any studies where this effect could bias the results and lead to wrong conclusions. In the case of the G. locorojo, our results suggest that it's reasonable to use light and medium tags at certain temperatures (in our case, from 24 to 28 °C), i.e., estimation of the distance travelled during the day at temperatures between 24 and 28 °C would not be biased by the use of light or medium tags.

What would be “negative effects” – can you quantify them so that we can determine when a tag is too heavy for an insect?

In the Discussion of the revised manuscript, we now consider the quantification (percentage decrease/increase) of tag effects on movement properties. Also, we replaced the tables that had p-values with tables of the calculated medians of movement properties. In our study, the degree of negative impact was dependent on temperature.

The general thoughts section should be integrated into an overall discussion. Right now it is very disjointed.

As we previously indicated, we have substantially revised the Discussion in order to make it clearer and more concise.

---

## [Decision Letter · Decision Letter 1]

23 Jun 2021

PONE-D-21-05023R1

Estimating the effect of tracking tag weight on insect movement using video analysis: a case study with an orthopteran

PLOS ONE

Dear Dr. Kaláb,

Thank you for submitting your manuscript to PLOS ONE. After careful consideration, we feel that it has merit but does not fully meet PLOS ONE’s publication criteria as it currently stands. Therefore, we invite you to submit a revised version of the manuscript that addresses the points raised during the review process.

Several issues remain to be addressed, and so I have asked for a revised manuscript in accordance with the suggestions of the Reviewers. The confounding of effects of time and temperature is particularly serious and may require additional experimentation to resolve. If the authors require more time to complete the task, the Editor would look favorably on granting an extension. 

We look forward to receiving your revised manuscript.

Kind regards,

Robert B. Srygley

Academic Editor

PLOS ONE

Reviewers' comments:

Reviewer's Responses to Questions

**Comments to the Author**

1. If the authors have adequately addressed your comments raised in a previous round of review and you feel that this manuscript is now acceptable for publication, you may indicate that here to bypass the “Comments to the Author” section, enter your conflict of interest statement in the “Confidential to Editor” section, and submit your "Accept" recommendation.

Reviewer #1: (No Response)

Reviewer #2: All comments have been addressed

2. Is the manuscript technically sound, and do the data support the conclusions?

Reviewer #1: No

Reviewer #2: Yes

3. Has the statistical analysis been performed appropriately and rigorously? 

Reviewer #1: Yes

Reviewer #2: I Don't Know

4. Have the authors made all data underlying the findings in their manuscript fully available?

Reviewer #1: (No Response)

Reviewer #2: Yes

5. Is the manuscript presented in an intelligible fashion and written in standard English?

Reviewer #1: No

Reviewer #2: Yes

6. Review Comments to the Author

Reviewer #1: You have not dealt with my primary concern about confounding temperature treatments with time in the arena. To be fair, I was not as clear as I could have been about describing the problem that I see. Based on the description of the experiment, each group of test animals is tested in the arena for 3 days in a row, first at the low temperature, then the medium temperature, then at the high temperature. So each group, on day three, has already been in the arena on two previous days. As such, time in the arena is confounded with temperature. I pointed out what I see as a potential solutions to the problem in my original review. You chose to just re-describe the methods. Given the way the paper is currently written I don’t believe you can distinguish the effects you attribute to temperature from effects due to days in the arena.

If you adjusted the p-values using the Benjamini and Hochberg method using something like the p.adjust R function and type = “BH”, then the citation should be Yekutieli & Benjamini (1999;

https://www.sciencedirect.com/science/article/abs/pii/S0378375899000415

The paper you cite does not adjust p-values, just describes how to adjust the acceptable p-value threshold.

Reviewer #2: Abstract: I am still missing the reason why temperature was included in this experiment; it is explained in the introduction, but should also be included in the abstract

Introduction: My concerns were addressed. Could benefit from some tighter language (i.e., make it a bit less wordy – get to the point faster), but no reason to reject the paper.

Line 6: rwith should with with

Line 10: what do you mean by units?

Methods: My concerns were addressed.

Results: Your response is appropriate and the tables with shading address my concerns.

Discussion: My concerns were addressed

Line 347: found should be find

7. PLOS authors have the option to publish the peer review history of their article (what does this mean?). If published, this will include your full peer review and any attached files.

Reviewer #1: No

Reviewer #2: No

---

## [Author Response · Author response to Decision Letter 1]

29 Jun 2021

Reviewer #1:

You have not dealt with my primary concern about confounding temperature treatments with time in the arena. To be fair, I was not as clear as I could have been about describing the problem that I see. Based on the description of the experiment, each group of test animals is tested in the arena for 3 days in a row, first at the low temperature, then the medium temperature, then at the high temperature. So each group, on day three, has already been in the arena on two previous days. As such, time in the arena is confounded with temperature. I pointed out what I see as a potential solutions to the problem in my original review. You chose to just re-describe the methods. Given the way the paper is currently written I don’t believe you can distinguish the effects you attribute to temperature from effects due to days in the arena.

We acknowledge that we possibly misunderstand the reviewer's primary concern, and thus our previous response may not be clear. However, as we stated in the previous response, we treated temperature as an independent variable, but we should emphasized in the response that each group was recorded in the same temperature for three days in row (e.g., in low temperature - group 1 was recorded three days in row, group 2 was recorded three days in row, …). We tried to describe this approach more clearly in a previous revision of the manuscript in section Experimental design, but it seems that our explanation is still not clear, so we alter the text, mainly editing the last paragraph of the subsection.

If you adjusted the p-values using the Benjamini and Hochberg method using something like the p.adjust R function and type = “BH”, then the citation should be Yekutieli & Benjamini (1999;

https://www.sciencedirect.com/science/article/abs/pii/S0378375899000415

The paper you cite does not adjust p-values, just describes how to adjust the acceptable p-value threshold.

Accepted. We used the “fdr” (“BH”) method in p.adjust function, and changed the reference to Yekutieli & Benjamini (1999) as suggested.

Reviewer #2:

Abstract: I am still missing the reason why temperature was included in this experiment; it is explained in the introduction, but should also be included in the abstract

We agree, thank you for your valuable comment. We have added this information also to the Abstract.

Introduction: My concerns were addressed. Could benefit from some tighter language (i.e., make it a bit less wordy – get to the point faster), but no reason to reject the paper.

The extensive introduction is the result of previous review edits, which heavily improved the readability and flow of this section. Although we reduced the first paragraph, the introduction expanded as we moved the blocks of text from other sections of the manuscript. We tried to be as concise as possible, but at the same time cover the basic information in the individual paragraphs in this order: tracking tag background - types and weights; pitfalls of tag attachments; context-dependent effects and temperature; methods of evaluation of the effects; our research - how differs from previous and what we want to answer. We consider the text in its current form to be balanced. We agree with the opponent that it is a bit wordy, but the topic is quite complex and we want to get the reader into the story in this way. 

Line 6: rwith should with with

fixed

Line 10: what do you mean by units?

replaced with “less than 1 mg”

Methods: My concerns were addressed.

Results: Your response is appropriate and the tables with shading address my concerns.

Discussion: My concerns were addressed

Line 347: found should be find

fixed

---

## [Editor Report · Decision Letter 2]

7 Jul 2021

PONE-D-21-05023R2

Estimating the effect of tracking tag weight on insect movement using video analysis: a case study with an orthopteran

PLOS ONE

Dear Dr. Kocarek,

Thank you for submitting your manuscript to PLOS ONE. After careful consideration, we feel that it has merit but does not fully meet PLOS ONE’s publication criteria as it currently stands. Therefore, we invite you to submit a revised version of the manuscript that addresses the points raised during the review process.

I appreciate your attention to the reviewers' comments and suggestions. I only have a few more suggestions and comments.

Title: Flightless needs to be added to the Title. Probably the best place is before 'orthopteran', as in 'with a flightless orthopteran'

Abstract: add 'flightless adult' before 'crickets Gryllus locorojo'

l. 96 change 'rises' to 'raised'

l. 206 a permanent repository, such as DRYAD, for your tracking application will need to be designated. Individual websites are not considered permanent.

l. 349 change 'significant affection' to 'significant effects on'

l. 372 you might also be interested in effects of weighting on dispersal of Anartia fatima doi: 10.3390/insects9030107

You do not mention the effect of changes in position of center of body mass, which is likely to be particularly important for tagged flying insects. 

l. 400 change 'affection of' to 'significant effects on'

l. 471 change 'brake' to 'break'

We look forward to receiving your revised manuscript.

Kind regards,

Robert B. Srygley

Academic Editor

PLOS ONE
---

## [Author Response · Author response to Decision Letter 2]

9 Jul 2021

TIitle: Flightless needs to be added to the Title. Probably the best place is before 'orthopteran', as in 'with a flightless orthopteran'

accepted

Abstract: add 'flightless adult' before 'crickets Gryllus locorojo'

accepted

l. 96 change 'rises' to 'raised'

fixed

l. 206 a permanent repository, such as DRYAD, for your tracking application will need to be designated. Individual websites are not considered permanent.

l. 349 change 'significant affection' to 'significant effects on'

fixed

l. 372 you might also be interested in effects of weighting on dispersal of Anartia fatima doi: 10.3390/insects9030107

You do not mention the effect of changes in position of center of body mass, which is likely to be particularly important for tagged flying insects. 

Thank you for this suggestion. We added this information to the end of the „limitations“ paragpraph, linking it with fact that our study is limited on flightless species, and with flying species there are possible pitfalls of tag attachment. We thing it fits in this paragraph better and keeps the consistence of Discussion (in proposed paragpraph we discuss only the weight effect in relation to our study).

l. 400 change 'affection of' to 'significant effects on'

accepted

l. 471 change 'brake' to 'break'

fixed

---

## [Editor Report · Decision Letter 3]

12 Jul 2021

Estimating the effect of tracking tag weight on insect movement using video analysis: a case study with a flightless orthopteran

PONE-D-21-05023R3

Dear Dr. Kocarek,

We’re pleased to inform you that your manuscript has been judged scientifically suitable for publication and will be formally accepted for publication once it meets all outstanding technical requirements.

Kind regards,

Robert B. Srygley

Academic Editor

PLOS ONE
---

## [Editor Report · Acceptance letter]

15 Jul 2021

PONE-D-21-05023R3 

Estimating the effect of tracking tag weight on insect movement using video analysis: a case study with a flightless orthopteran  

Dear Dr. Kocarek:

I'm pleased to inform you that your manuscript has been deemed suitable for publication in PLOS ONE. Congratulations! Your manuscript is now with our production department. 

Kind regards, 

on behalf of

Dr. Robert B. Srygley 

Academic Editor

PLOS ONE